# An integrated dataset of daily lake surface water temperature over Tibetan Plateau

Linan Guo[1,2], Hongxing Zheng[3], Yanhong Wu[1,4]*, Lanxin Fan[1,4], Mengxuan Wen[1], Junsheng Li[1,4], Fangfang Zhang[1], Liping Zhu[2], Bing Zhang[1,4]*

[1]Key Laboratory of Remote Sensing Science, Aerospace Information Research Institute, Chinese Academy of Sciences, Beijing 100094, China
[2]Institute of Tibetan Plateau Research, Chinese Academy of Sciences, Beijing 100101, China
[3]CSIRO Land and Water, Canberra, ACT 2601, Australia
[4]University of the Chinese Academy of Sciences, Beijing 100049, China

*Correspondence to*: Yanhong Wu (wuyh@radi.ac.cn); Bing Zhang (zb@radi.ac.cn)

**Abstract.** Lake surface water temperature (LSWT) is a critical physical property of the aquatic ecosystem and an evident indicator of climate change. By combining the strengths of satellite-based observation and modelling, we have produced an integrated daily lake surface water temperature for 160 lakes across the Tibetan Plateau where *in-situ* observation is limited. The MODIS-based lake-wide mean LSWT in the integrated dataset includes that for the daytime, night-time and for the daily mean for the period 2000–2017. The MODIS-based daily mean LSWT is used to calibrate a simplified physically based model (i.e., modified *air2water* model), upon which a complete and consistent daily LSWT dataset is reconstructed for the period 1978–2017. The reconstructed LSWT dataset is validated by comparing against both the satellite-based and in-situ observations. The validation shows that the reconstructed LSWT is in good agreement with the observations. According to the reconstructed LSWT dataset, it is found that annual LSWT of lakes in the Tibetan Plateau has increased significantly in the period 1978–2017 with increase rate ranging at 0.01 to 0.47 ℃ 10a$^{-1}$. The warming rate is higher in winter than in summer. The integrated dataset is unique for its relatively large tempo-spatial span (1978-2017) and high temporal resolution. The dataset together with the methods developed can contribute to the research community in exploring water and heat balance changes and the consequent ecological effects at the Tibetan. Data from this study are openly available via the Zenodo portal, with DOI https://doi.org/10.5281/zenodo.6637526 (Guo et al., 2022).

## 1 Introduction

Lake surface water temperature (LSWT) is a critical physical property of the aquatic ecosystem and an evident indicator of climate change (Austin and Colman, 2008; Livingstone, 2003; Williamson et al., 2009). Rapid rise in water temperature has been observed in many lakes around the world, which not only reflects the changes in lake heat budget associated with global warming (Bates et al., 2008;Dokulil, 2014) but also has resulted at a succession of changes in physical, chemical, and biological processes within the lake system (Hondzo and Stefan, 1993; Ke and Song, 2014; Naumenko et al., 2006; Ngai et al., 2013;

Rahel and Olden, 2008; Schindler, 2001; Woolway and Merchant, 2017; Liu et al., 2021). One of the noticeable consequences of changes in lake water temperature is the considerable change in lake ice phenology (e.g., the freezing-up and breaking-up dates of lake ice) found at mid- to high-latitude or high-altitude regions around the world during the past decades (Livingstone, 1997;Takacs et al., 2018;Tian et al., 2015; Prowse et al., 2011).

Lake water temperature records based on *in-situ* measurements are not widely available due to its costs and the geographical restrictions of the TP. However, in the Tibetan Plateau (TP), where there are more than 1,100 alpine lakes with area larger than 1 km$^2$ and elevation above 4000 m, most of the lakes have no *in-situ* water temperature records due to its harsh nature for ground observation (Zhang et al., 2014). Satellite-based observation has been showing highly powerful in providing continuous worldwide records of lake surface temperatures and has developed rapidly in recent decades with increasingly higher temporal and spatial resolutions (Liu et al., 2019; Prats et al., 2018; Schneider and Hook, 2010). At global scale, currently accessible satellite-based lake surface temperature datasets include that from the ARC-Lake (ATSR Reprocessing for Climate: LSWT & Ice Cover) (Layden et al., 2015) and the GLTC (Global Lake Temperature Collaboration) (Sharma et al., 2015). The global datasets are highly valuable in stimulating research on inland water bodies (e.g., Moukomla and Blanken, 2017; Piccolroaz et al., 2020; Torbick et al., 2016; Zhang et al., 2014). The datasets however are limited for use in the Tibetan Plateau as they cover only a handful of lakes within the TP region and are temporally incomplete (Liu et al., 2019). Two other surface water temperature datasets for lakes in the Tibetan Plateau have recently been produced by Wan et al. (2017) and Liu et al. (2019) using AVHRR and MODIS respectively. One of the datasets bases on MODIS land surface temperature products and provides 8-day mean surface temperature of 374 lakes for the period 2001-2015 (Wan et al., 2017), while the other bases on AVHRR and presents daytime lake surface water temperature of 97 lakes with area above 80km$^2$ for the period 1981-2015 (Liu et al., 2019). Both the two datasets for lakes in the Tibetan Plateau, however, have quite a few missing data caused by revisit period of satellite or inconsistency in the time series due to calibration among the successive satellites.

Lake surface water temperature can alternatively be derived or reconstructed basing on well calibrated modelling (Layden et al., 2016; Prats and Danis, 2019). Process-based numerical models have been used widely to investigate thermodynamics of lakes at local scales (Goudsmit et al., 2002; Kirillin et al., 2017; Launiainen and Cheng, 1998; Peeters et al., 2002; Stepanenko et al., 2016), which usually requires detailed over-lake meteorological data (e.g., wind speed, humidity, cloud cover) as inputs and *in-situ* measurements to calibrate and validate the models (Bruce et al., 2018). For regions with scarce in-situ meteorological observations, simple statistical models (Mccombie, 1959; Webb, 1974; Sharma et al., 2008) or simplified physically based model (Piccolroaz et al., 2013) have been developed for use, which may require only air temperature as input. The air2water model proposed by Piccolroaz et al. (2013) is a hybrid model with strong physical base but simplifies the thermodynamic equations to minimize the input requirement while preserving the robustness of deterministic models. The air2water has shown capable in providing similar performance, in terms of simulating LSWT, to process-based models, and to be an effective tool in reconstructing historical LSWT (Czernecki and Ptak 2018; Piccolroaz et al., 2020; Schmid and Köster 2016) and in investigating LSWT responses to climate change for lakes with different morphological characteristics around

the world (Piccolroaz et al., 2015; Prats and Danis, 2019; Toffolon et al., 2014). The simplified model is especially competitive and practicable for regions with limited *in-situ* observations to drive and calibrate a more complex process-based model.

We herein combine the strengths of both the remotely sensed and model-based approach in producing an integrated dataset on daily surface water temperature of 160 large lakes across the Tibetan Plateau for the period 1978–2017. For the combination, the remotely sensed LSWT is used to calibrate and validate the air2water model, which is then applied to reconstruct a complete

and consistent daily LSWT time series for each studied lake and extend LSWT dataset to a wider time span. The integrated dataset demonstrates potentials in investigating variabilities and changes in LSWT across the Tibetan Plateau during the past decades. It could be valuable for assessing the impacts of climate warming on the dynamics of water and heat budget, water quality and aquatic biota in lakes across the Tibetan Plateau.

## 2 Study area

The Tibetan Plateau locates between 26°00′–39°47′N and 73°19′–104°47′E with mean elevation over 4500 m and area about $2.5 \times 10^6$ km$^2$. It is known as the "Asian water tower", contributing to most major rivers in Asia (Wu and Lei, 2014). The total area of lakes in the plateau is around 45,000 km$^2$, most of which are located at the altitudes between 4000 m and 5000 m (Yao et al., 2015). Our dataset includes 160 lakes in the Tibetan Plateau covering most major lakes with area above 40 km$^2$ (Fig.1). All lakes were selected from Records of Lakes in China (Wang and Dou, 1998). The general properties of the studied lakes

are listed in the lake_info.csv file of the dataset, which includes the names, locations and areas of the lakes.

## 3 Methods

Fig. 2 shows the overall framework of our study in integrating lake surface water temperature dataset for the largest lakes across the Tibetan Plateau. The MOD11A1 product is the source data for the satellite-based LSWT, while the air temperature data from meteorological station is the source data used to drive the air2water model. The satellite-based observation is used

to calibrate and validate the air2water model. The details of the procedures and methods are described in the following sections.

### 3.1 Satellite-based observation of lake surface water temperature

The first step of our effort in producing the integrated LSWT dataset is to derive the satellite-based observation from the MOD11 (version 6) provided by NASA's Earth Observing System Data and Information System (EOSDIS, https://earthdata.nasa.gov). The MOD11 (Wan, 2013) is the land surface temperature and emissivity products retrieved at 1km

pixels by the generalized split-window algorithm and 6km grids by the day/night algorithm mainly based on bands 31 (10.78–11.28μm) and 32 (11.77–12.27μm) of MODIS (Moderate-resolution imaging spectroradiometer) aboard the Terra and Aqua satellite, where Terra overpasses the equator at around 10:30 A.M. (10:30 P.M.) local time and Aqua at around 1:30 A.M. (1:30 P.M.). The MOD11 includes products with temporal resolutions of daily, 8-day, and monthly respectively. Validation of

the MODIS LST product for water surface temperatures against in-situ measurements has been conducted and the absolute

differences have been reported to be within the range of 0.8–1.9 °C (Crosman and Horel, 2009, 2008; Hulley et al., 2011; Reinart and Reinhold). The bias of MODIS-based LSWT was reported to be around -1.74°C (Song et al., 2016) and -1.4°C (Zhang et al., 2014) over the TP when compared to the limited in-situ observations. Different from the 8-day dataset by Wan et al. (2017), herein, we used MOD11A1 (Terra product of Land Surface Temperature/Emissivity Daily L3 Global 1km) instead of MOD11A2 to produce the daily lake surface temperature dataset. The daily product is used herein also because it is

more suitable to calibrate and validate the air2water model at the same temporal resolution. The MOD11A1 is a tile of daily Level 3 product at 1km spatial resolution corresponding to the earth locations on the sinusoidal projection.

In our dataset, the satellite observation of LSWT for a specific lake is the lake-wide mean temperature of all 1-km pixels within the lake, which is different to the temperature of the centroid pixels presented by Zhang et al., (2014) and is more consistent with our model settings. In calculating the lake-wide mean surface water temperature, pixels within a lake are identified by the

boundaries of the lakes (shapefiles included in the dataset), which are mainly from National Tibetan Plateau Data Center (TPDC, http://data.tpdc.ac.cn) and cross-checked by lake boundary maps from HydroLAKES (http://www.hydrosheds.org/) (Messager et al., 2016) and Google Earth. To remove the boundary effects (i.e., where pixels could be mix of land and water), the boundaries are buffered inward by 1km. The daily surface water temperature is the mean of daytime and night-time LST in the MOD11A1. All the data processing procedures are conducted via Google Earth Engine (GEE). In addition, the

MOD11A1 product has noticeable missing values because of cloud effects and the limitation of the algorithm. The gaps in our MODIS LSWT series were marked as "-999".

**3.2 Reconstructing lake surface water temperature**

The satellite-based lake surface water temperature is accessible since 2000 and with missing data due to the limitation in the MODIS. To investigate the long-term changes in surface water temperature for lakes in the Tibetan Plateau and their responses

to climate change, in our integrated datasets, we used a slightly modified air2water model to reconstruct the daily lake surface water temperature for the period 1978-2017 based on daily air temperature data from Dataset of daily climate data from Chinese surface stations (V3.0) provided by China Meteorological Data Service Centre of National Meteorological Information Centre (http://data.cma.cn/data/cdcdetail/dataCode/SURF_CLI_CHN_MUL_DAY_V3.0.html).

The air2water (Piccolroaz et al., 2013) is a semi-physical model designed to simulate lake surface temperature principally

based on lake surface heat balance and can be expressed as:

$$\rho c_p V_s \frac{dT_w}{dt} = A H_{net}, \tag{1}$$

where $T_w$ is surface water temperature, $H_{net}$ is the net heat flux per unit surface, $A$ is the surface area of the lake, $V_s$ is the volume of water involved in the heat exchange with the atmosphere, $\rho$ and $c_p$ is the density and specific heat capacity of water. The model considers all contributions to the heat balance ($H_{net}$), which are represented as a function of air temperature or are

parameterized (Piccolroaz et al., 2013;Piccolroaz, 2016). The original air2water model has simplified the lake heat balance by introducing eight calibratable parameters $a_{1-8}$ with air temperature as the only required input, which can be expressed as:

$$\frac{dT_w}{dt} = \frac{1}{\delta}\left\{a_1 + a_2 T_a - a_3 T_w + a_5 \cos\left[2\pi\left(\frac{t}{t_y} - a_6\right)\right]\right\}, \tag{2}$$

$$\delta = \begin{cases} exp\left(-\frac{T_w - T_h}{a_4}\right), & for\ T_w \geq T_h \\ exp\left(-\frac{T_h - T_w}{a_7}\right) + exp\left(-\frac{T_w}{a_8}\right), & for\ T_w \leq T_h \end{cases}, \tag{3}$$

where, $T_a$ is air temperature, $t_y$ is days of a year. $\delta = D/D_r$ is normalized well-mixed depth, where $D$ is the depth of well-mixed surface layer (the epilimnion thickness), $D_r$ is the maximum thickness. $T_h$ is the deep water temperature with default value being 4℃. The $a_{1-4}$ in Eq. (2) are the most sensitive parameters determining model performance (Piccolroaz et al., 2013).

The original air2water is limited to simulate the surface temperature of open water, which results in difficulties for applications in lakes with long ice cover duration. For this sake, we assume that when the lake is completely covered by ice, the heat exchange between air and water is blocked and surface energy balance becomes:

$$\frac{dT_i}{dt} = a_9 + a_{10} T_a - a_{11} T_w + a_{12} \cos\left[2\pi\left(\frac{t}{t_y} - a_{13}\right)\right], \tag{4}$$

where $T_i$ is the ice surface temperature, $a_{9-13}$ have similar physical significance to $a_{1,2,3,5,6}$. To represent the state-shift of lake between open water and ice-covered, two additional parameters $a_{14}$ and $a_{15}$ are introduced to determine the states. The lake surface temperature $T_L$ can be expressed as:

$$T_L = \begin{cases} T_w, & for\ T_L \geq a_{15} \\ T_i, & for\ T_L \leq a_{14}, \\ (1 - K_{ice})T_w + K_{ice}T_i, & for\ a_{15} > T_L > a_{14} \end{cases} \tag{5}$$

Where, $K_{ice} = \sqrt{(a_{15} - T_L)/(a_{15} - a_{14})}$ is the proportion of ice on the surface of the lake. Same as the original model, the model equations were solved numerically by using the Crank-Nicolson numerical scheme at daily time step.

The daily air temperature used to drive the modified air2water model are from 31 meteorological stations in the Tibetan Plateau (Fig.1). The air temperature above each lake is interpolated from the nearest station and adjusted by a lapse rate of 0.65℃ 100m$^{-1}$ (Hu et al., 2014). The model is calibrated against the derived remotely sensed LSWT described in Section 3.1 using the PSO (Particle Swarm Optimization) approach (Kennedy and Eberhart, 1995; Piccolroaz, 2016). The objective function of model calibration is the widely adopted Nash–Sutcliffe efficiency coefficient (NSE) (Nash and Sutcliffe, 1970). To ensure the rationality of the calibrated model parameters, a physically consistent priori range are assigned to each parameter (Piccolroaz, 2016) and $D_r$ is bounded by the average depth of the lake obtained from Records of Lakes in China (Wang and Dou, 1998) and HydroLAKES (Messager et al., 2016). The calibration period is set to be the period 2000-2012, while 2013-2017 is considered as the validation period. It should be noticed that the satellite-derived water temperature is measurement of instantaneous water temperature at the top of the surface (~10– 20μm deep), known as skin temperature. The skin temperatures can differ from surface water temperature because the thermal structure of the first metres of the water column is not uniform

under all conditions. Nevertheless, satellite-derived water temperature (skin temperatures) are relevant and sufficient for the

calibrating and validating hydrodynamic or water quality models of lakes (Andréassian et al., 2012; Prats and Danis, 2017; Prats et al., 2018). For comparison and further validation, the dataset produced by Liu et al. (2019) based on AVHRR was also used.

## 4 Results and discussions

### 4.1 MODIS-based LSWT of lakes across the Tibetan Plateau

Three subsets of lake-wide mean surface water temperature are derived and included in our integrated dataset, which are daily daytime LSWT, daily night-time LSWT and daily mean LSWT for the period 2000-2017. The lake surface water temperature derived from the MOD11A1 is compared against two satellite-based datasets released by other researches, which were based on AVHRR (Liu et al., 2019) and ARC-Lake (Layden et al., 2015) respectively.

Fig.3 shows the comparison between our MOD11A1-based LSWT against the AVHRR-based LSWT (i.e., the TPlake_Temp

dataset by Liu et al., 2019) for 77 sharing lakes. The TPlake_Temp provides daily LSWT for the period 1981–2015 and for 97 lakes in the TP, which however is with different length of data gaps among the lakes. It should be noted that the LSWT from the TPlake_Temp is that of the daytime instead of the daily mean, hence, it is compared against our MODIS-based daytime LSWT instead of mean daily LSWT herein. It is found that the LSWT from the two sensors (i.e., AVHRR and MODIS) are largely comparable with each other for the studied lakes, where the NSE and $R^2$ between the LSWT from the two datasets is

higher than 0.7 for more than 80% of the lakes, and the bias ranges from 0.6 to 3.38℃. It is worth mentioning that the MODIS-based LSWT would be better for investigating the long-term changes in lake surface temperature in the Tibetan Plateau since the AVHRR-based LSWT from the TPlake_Temp dataset has more missing data and was reported having abrupt shifts in LSWT due to inconsistent calibration among the successive satellites (Liu et al., 2019).

We further compare the MODIS-based LSWT against that from ARC-Lake. The ARC-Lake dataset (version 3) included

ATSR-2/AATSR-based lake surface temperatures for the period 1995-2012 for 1,628 target water bodies distributed globally.. Here the LSWTs of daytime and night-time in ARC-Lake are averaged to present daily mean LSWT. Since the ARC-Lake LSWT product is only for open water and tag temperature frozen season with value 0 ℃, in the comparison, only the period when both MODIS-based and ARC-Lake LSWT are above 0℃ is considered. LSWTs of 11 studied lakes (location shows in Fig. S1 in Supplementary) are available from the ARC-Lake for the comparison. Besides some outliers where the difference

between the two satellite observations is larger than 5 ℃ (around 8 % of the observations in the studied periods), as shown in Fig. 4, the two satellite-based observations of LSWT are mostly comparable with NSE>0.7 and bias ranging at -1.23 ℃ to 0.94 ℃.

## 4.2 Reconstructed LSWT by air2water

The reconstructed LSWT by the slightly modified air2water model is compared and validated by the MODIS-based LSWT.
As shown in Fig.5a and 5b, the spatial pattern of the long-term mean annual lake surface water temperature from the
reconstructed LSWT is close to that based on MOD11A1 with overall NSE=0.977 and $R^2$=0.987. For the period 2000–2017,
NSE and $R^2$ of most lakes (>90%) are both above 0.8 with bias ranging at ±0.55°C. The bias for the validation period (2013–
2017) is slightly higher than that for the calibration period (see Fig. S2 in the Supplementary) range from -1.7 to 0.9 °C.
Although there is a tendency showing that NSE is relatively higher for lakes with lower altitude and higher latitude, model
performance is found weakly related to altitude and latitude of the lakes (see Fig.S3 in the Supplementary). The results indicate
there is no substantial systematic bias in the reconstructed LSWT. The modified air2water model hence is reliable in
reconstructing LSWT for lakes at different altitude or latitude zones. However, it should be noticed that uncertainties exist in
the reconstructed LSWT due to the uncertainties not only in the calibrated model but also in the air temperature inputted to the
model. Summaries of model performance for each month presented in Fig. S4 (in the Supplementary) show that the simulated
LSWT is with smaller bias in July and November. As mentioned in Section 3.2, the daily air temperature above the lakes is
that interpolated from the nearest meteorological stations with elevation adjustment, where the performance of the interpolation
would be affected by the density and locations of the stations. Currently, the available meteorological stations are sparsely
located at the western TP (Fig.1), which may result at inherent uncertainties of the reconstructed LSWT for lakes in the western
TP. As shown in Fig.5c and 5d, though there is no significant difference in model performance with respect to the locations of
the lakes, it is worth exploring further in the future the effects of the interpolation approaches on the simulation of LSWT.

The *in-situ* observed LSWT is not widely available for lakes in the Tibetan Plateau to conduct an overall validation of the
modelling results. Nevertheless, we have compared the modelling results against the best publicly available *in-situ* surface
water temperature data for lakes in the Tibetan Plateau (Table S1, Fig.S6 and Fig.S7 in Supplementary), which include
sequential observation of 4 lakes (i.e., the Ngoring Lake, Serling Co, Dogze Co, Bangong Co) and sporadic observation
(simulated lake surface water temperature of day same to the observation) of 41 lakes. As shown in Fig.6, the simulated lake
surface temperature is in good agreement temporally with the sequential observations ($R^2$=0.97, 0.92, 0.90, 0.97 for Ngoring
Lake, Serling Co, Dogze Co, Bangong Co respectively) and spatially with the sporadic observation ($R^2$=0.94). Compared
against the in-situ observations, the RMSE of the simulated temperature is around 2.0 °C for the 45 lakes listed in Table S1.
It is noticed that the bias in the simulation mainly due to its underestimation for the warmer seasons. Taking Ngoring Lake for
example, for the season when the in-situ observed temperature is above 10°C, the RMSE could reach to 2.42 °C though $R^2$ is
higher than 0.70. The bias of the simulated water temperature could be reduced or corrected if the model is calibrated against
the observations. However, it is important to note that the simulated water temperature is not completely equivalent to the in-
situ observations. This is because the simulations represent the lake-wide mean temperature of the skin layer, while the in-situ
observation used herein is the profile mean temperature for a fixed location in the lake.

## 4.3 Long-term trends of LSWT in the Tibetan Plateau

On the basis of the reconstructed daily LSWT dataset, the long-term changes of surface water temperatures of lakes across the Tibetan Plateau are detected by using the Mann-Kendall trend analysis approach (Kendall, 1955; Mann, 1945). The daily and annual variation of a sample lake is shown in Fig. S5 (Supplementary). Fig.7 shows that, for the period 1978–2017, the annual LSWT of most lakes (except for Lake Beidao and Lake Changhong) increase significantly at rates ranging from 0.01 to 0.47 ℃ 10 a-1, which is consistent with but comparingly smaller than that of the increasing rate in air temperature (0.3-0.9 ℃ 10 a-1) indicating the contributions of heat storage capacity of the lakes. Lakes in the southern TP are found generally with higher warming rate than those in the northern TP. The increase in LSWT is more evident in the winter season (December to February) than in the summer season (June to August). Except for Lake Kuhai, the LSWTs of all the other lakes increase significantly at a rate ranging at 0.06 and 0.96 ℃ $10a^{-1}$. In summer, while most of the lakes (125 out of 160) show significantly increase in LSWT, the LSWTs of 13 lakes locating at the northern part of the TP decrease insignificantly. It should be noted that the warming and cooling trends could inherit uncertainties from the modelling, which needs further validation when more in-situ observations becoming available.

## 5 Data availability

Table 1 gives the details of the data included in our integrated dataset. In the dataset, the properties of the lakes (including name, latitude, longitude, altitude) are listed in the lake_info.csv file. The boundaries of the lakes are provided in the ESRI shapefile format. The satellite-based observations of the LSWT are given as the MOD11A1_DT and MOD11A1_NT for the daytime and night-time lake-wide mean lake surface water temperature, while the MOD11A1_DM is the mean of them representing the daily mean surface water temperature. The time span of the MODIS-based LSWT is for the period 2000–2017. The Air2Water_DM is the reconstructed daily LSWT based on the modified air2water model for the period 1978–2017. Time series plot for each lake are also included in the dataset for readers to have a quick view. The dataset is archived and openly accessible via the Zenodo portal: https://doi.org/10.5281/zenodo.6637526 (Guo et al., 2022). The GEE scripts are available at: https://code.earthengine.google.com/563820c56b30595de901c21aef5f0c71.

## 6 Code availability

The source codes of the modified model is open source at GitHub: https://github.com/Siyu1993/ModifiedModel.git.

## 7 Conclusions

An integrated daily lake surface water temperature has been produced for 160 lakes across the Tibetan Plateau by combining the strengths of satellite-based observation and model-based approaches. The satellite-based lake-wide mean LSWT is derived

from MOD11A1 via Google Earth Engine, which includes that for the daytime, night-time and for the daily mean for the period 2000–2017. The dataset is found comparable with other satellite-based LSWT products (e.g., LSWT from AVHRR and

ARC-Lake) but unique due to its tempo-spatial span and resolution. The satellite-based LSWT enables the calibration and validation of a simplified heat balance model (i.e., air2water) at regions with scarce ground-based observation (like the Tibetan Plateau).

The modified *air2water* model is found successfully in reconstructing the daily LSWT of lakes across the Tibetan Plateau by extending the LSWT time series to the period 1978–2017 and filling the time gaps in the satellite-based LSWT though with

uncertainties from model inputs and model parameterization. The completeness and consistency of the reconstructed LSWT therefore is reliable and valuable to investigate the long-term variation and changes of LSWT in the Tibetan Plateau. According to the reconstructed LSWT dataset, the annual LSWT is found increased significantly in the period 1978–2017 with increase rate ranging at 0.01 to 0.47 ℃ $10a^{-1}$. The warming trend of the lakes is more evident in winter than in summer. The integrated dataset together with the methods introduced herein can contribute to the research community in exploring water and heat

balance changes in the Tibetan Plateau and the consequent ecological effects in the future researches.

**Author contributions.** YW, HZ, BZ and LG conceived the research. LG, YW and HZ developed the approaches and datasets. LG, MW and LF collected basic data of lakes. JL, SW checked the results. LG, HZ and YW wrote the original draft. BZ and LZ revised the draft.

**Competing interests.** The authors declare that they have no conflict of interest.

**Acknowledgements.** We gratefully acknowledge the Climate Data Center, National Meteorological Information Center, China Meteorological Administration, for providing the long-term meteorological data of the 31 stations.

**Financial support.** This research has been supported by the Second Tibetan Plateau Scientific Expedition and Research Program (STEP) (Grant No. 2019QZKK0202) and National Natural Science Foundation of China (Grant No. 41671203).

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

**Table 1 Data descriptions.**

|  | Data | Description | Period |
|---|---|---|---|
|  | **Lake boundary** | Boundaries of lakes in the format of ESRI shapefile | - |
|  | **LakeInfo** | Properties of lakes in the format of csv | - |
| **Daily LSWT** | MOD11A1_DT | Lake surface temperature derived from MOD11A1 daytime products. | 2000–2017 |
|  | MOD11A1_NT | Lake surface temperature derived from MOD11A1 nighttime products. | 2000–2017 |
|  | MOD11A1_DM | Mean of MOD11A1_DT and MOD11A1_NT. | 2000–2017 |
|  | Air2Water_DM | Model-based daily lake surface water temperature. | 1978–2017 |


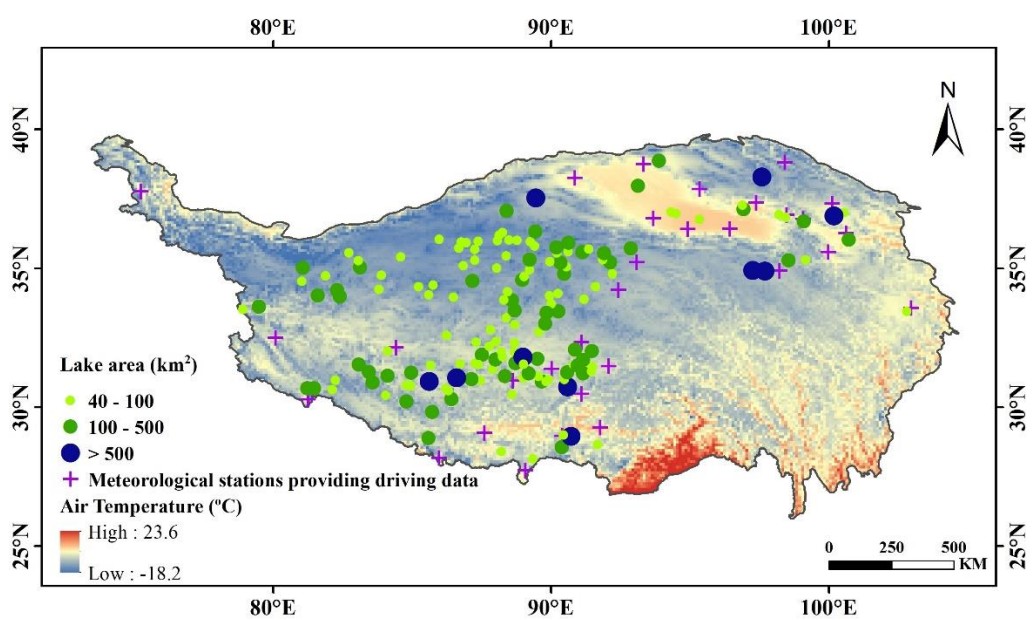

**Figure 1: Locations of studied lakes in the Tibetan Plateau.**

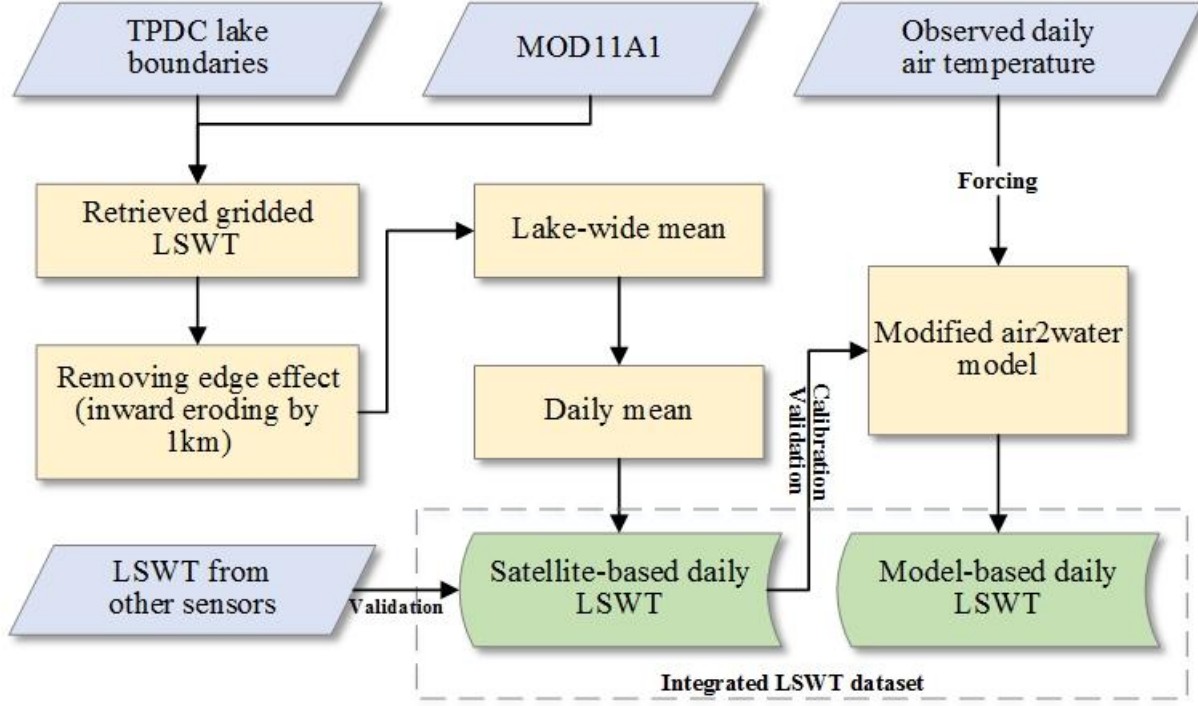

**Figure 2: Framework for integrating lake surface water temperature data.**

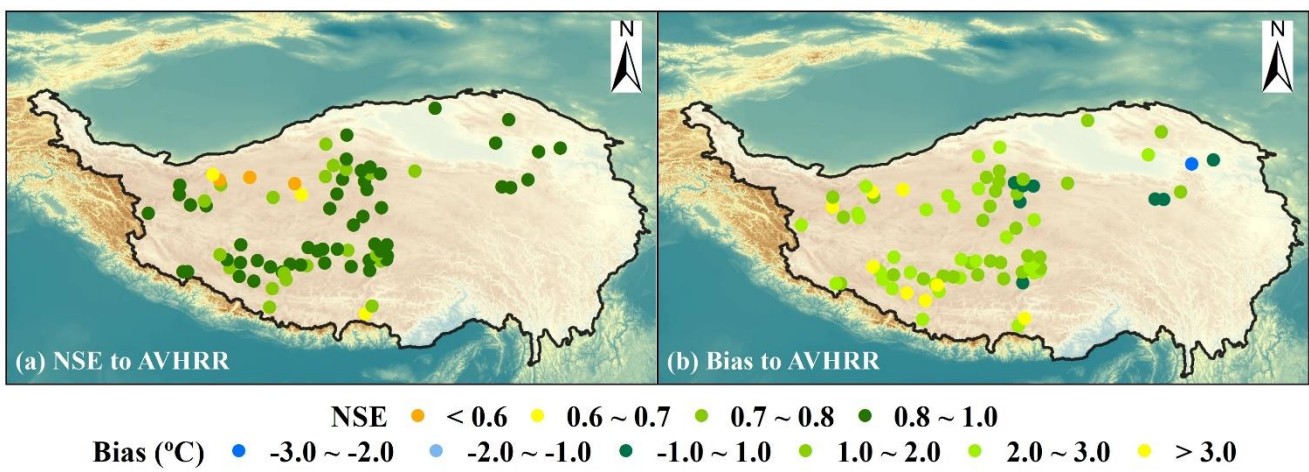

**Figure 3: Comparison of LSWT from MOD11A1 and AVHRR LSWT for 77 lakes evaluated by NSE (a) and bias (b).**

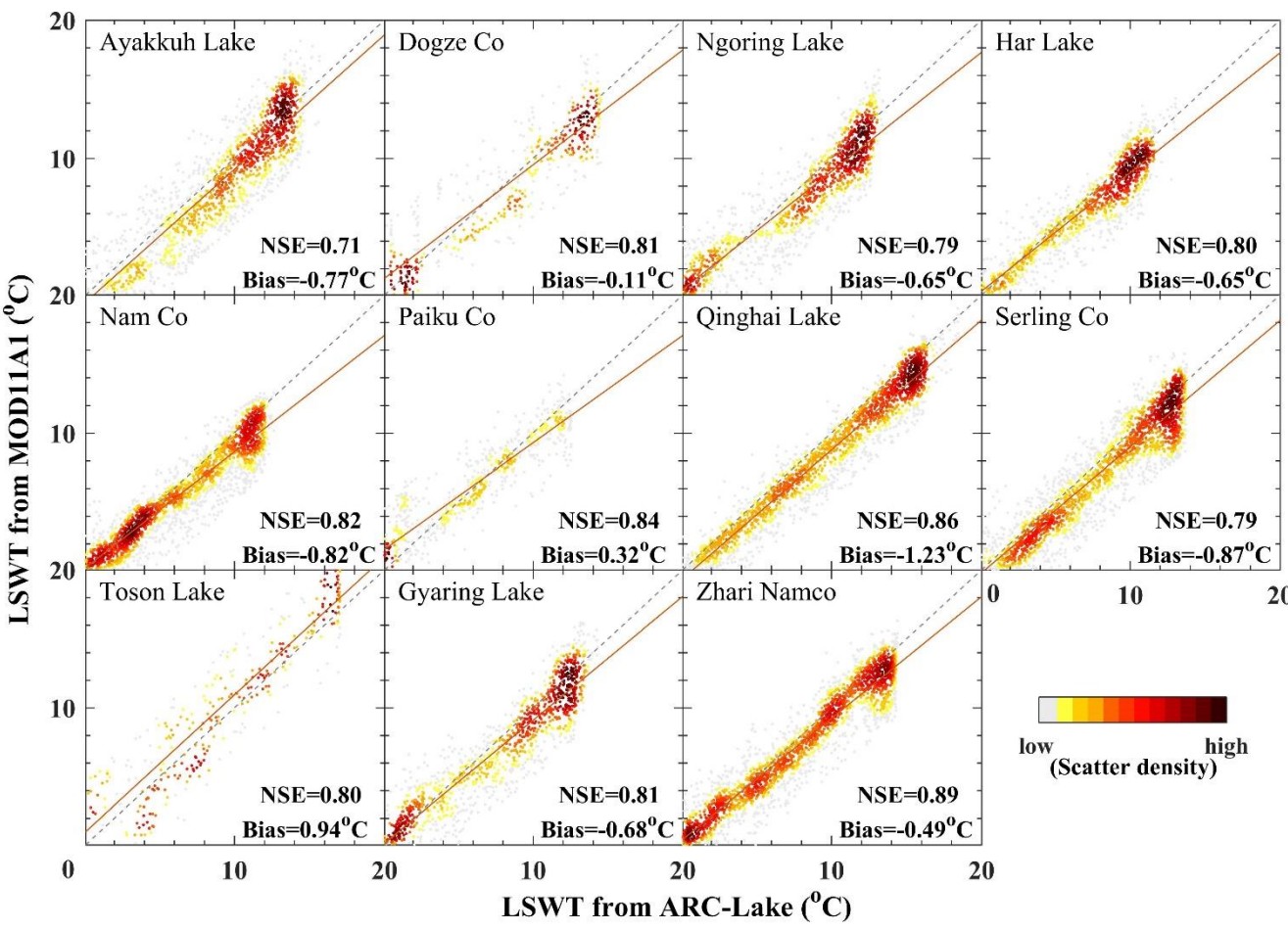


**Figure 4: Comparison of LSWT from MOD11A1 and ARC-Lake for the period with LSWT>0°C. The dash line is the 1:1 line, the solid line is the linear regression line.**

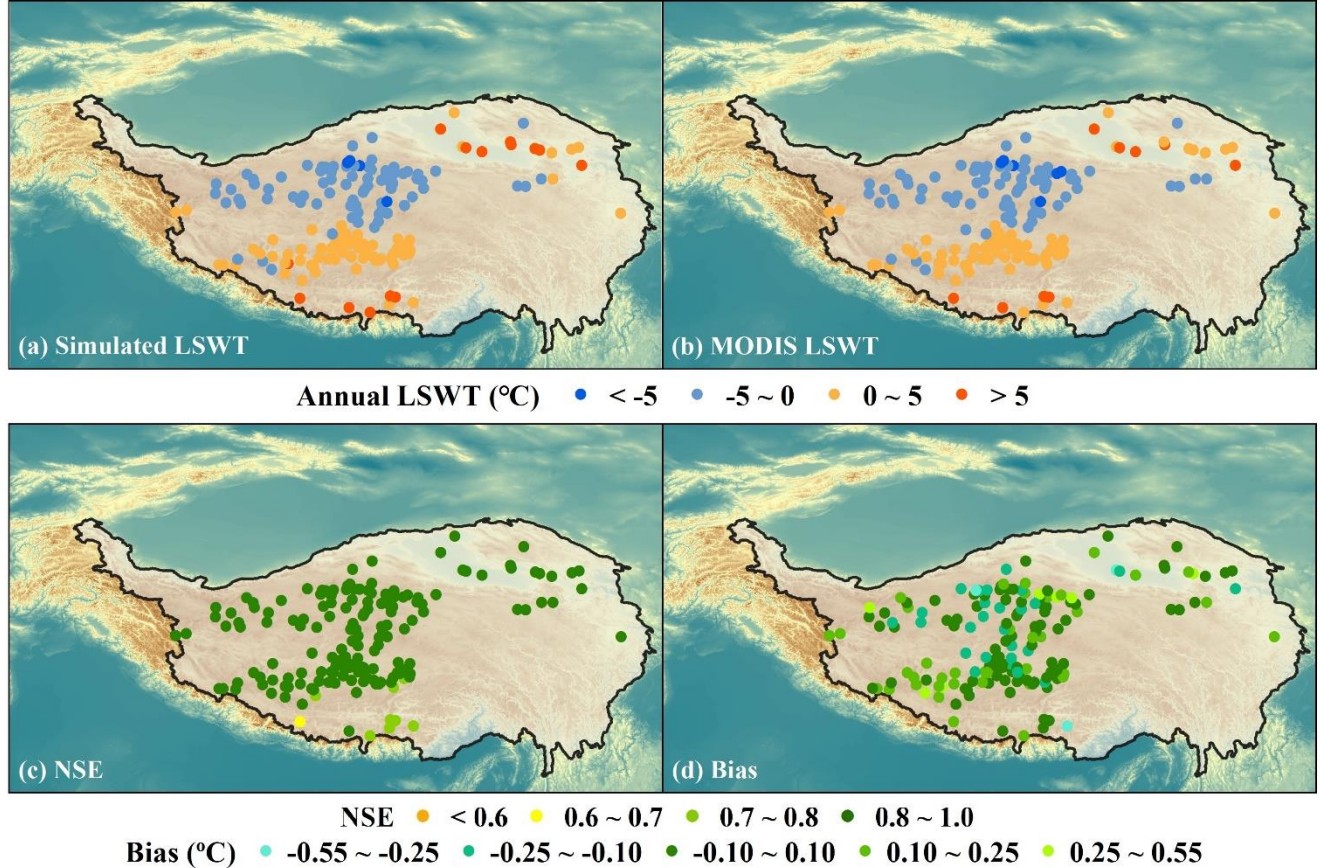

**Figure 5: Spatial comparison between MOD11A1-based (a) and reconstructed (b) annual mean LSWT and spatial distribution of model performance evaluated by NSE (c) and bias (d).**

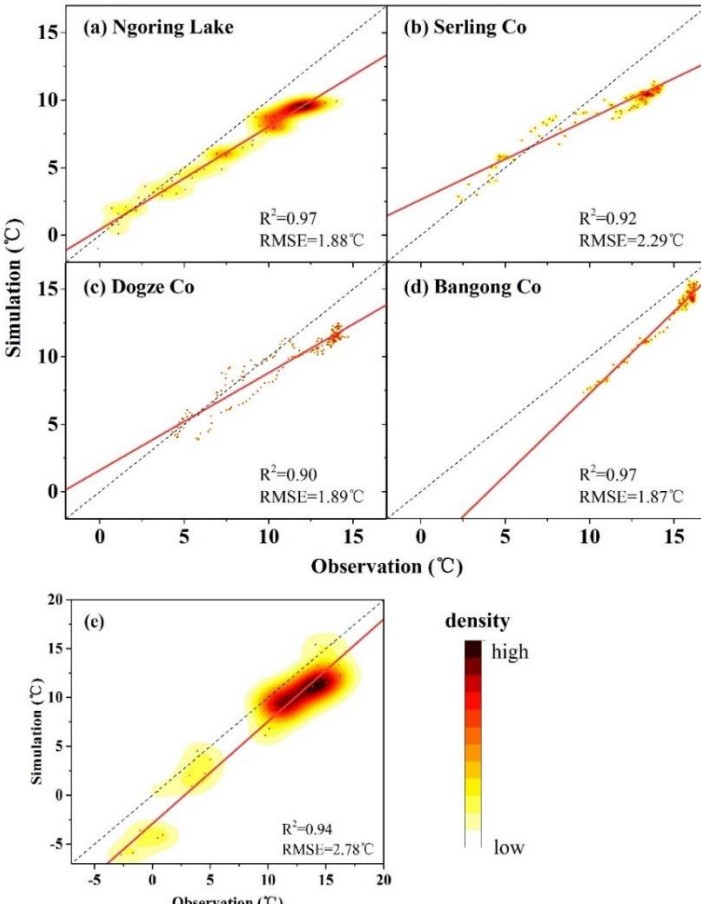

**Figure 6 Validation of modeling results against *in-situ* observation summarized in Table S1: (a)-(d) is the temporal validation respectively for the 4 lakes with sequential observation; (e) is the spatial validation for the 41 lakes with sporadic observation. The solid lines are the regression lines. The dashed lines are the 1:1 lines.**

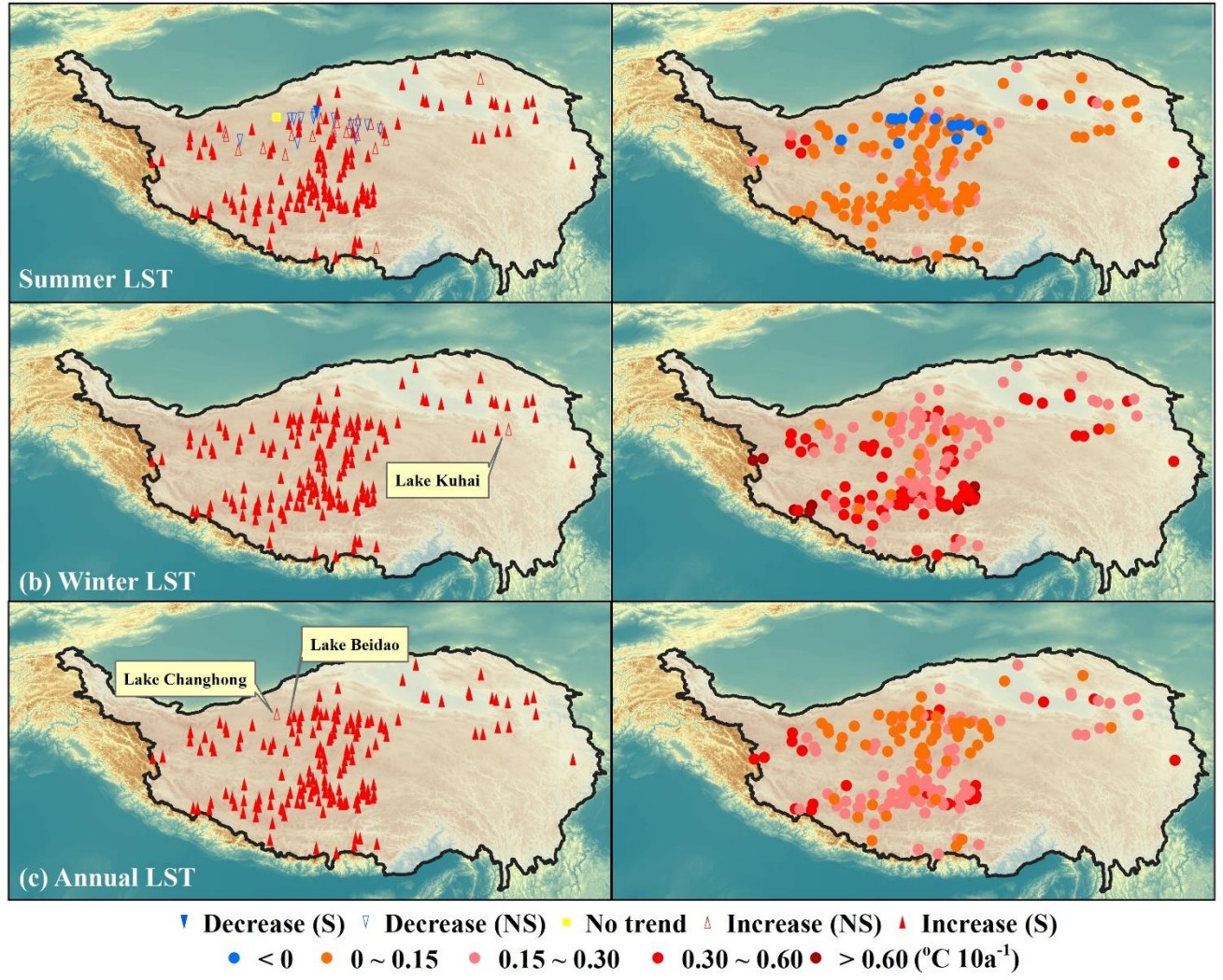

**Figure 7: Long-term trends of summer (Jun-Aug), winter (Dec-Feb) and annual LSWT during 1978–2017.**
