# Peer review of "An integrated dataset of daily lake surface water temperature over Tibetan Plateau"

_Earth System Science Data, 2021_

## Author Comment (AC1)

**An integrated dataset of daily lake surface water temperature over Tibetan Plateau**

Linan Guo[1,2], Hongxing Zheng[3], Yanhong Wu[1,2]*, Lanxin Fan[1,2], Mengxuan Wen[1], Junsheng Li[1,2], Fangfang Zhang[1], Liping Zhu[4], Bing Zhang[1,2]*

[1]Key Laboratory of Remote Sensing Science, Aerospace Information Research Institute, Chinese Academy of Sciences, Beijing 100094, China

[2]University of the Chinese Academy of Sciences, Beijing 100049, China

[3]CSIRO Land and Water, Canberra, ACT 2601, Australia

[4]Institute of Tibetan Plateau Research, Chinese Academy of Sciences, Beijing 100101, China

Table S1 Summary of *in-situ* lake surface temperature observation used for validation. The first 4 lakes denoted by * are with relative long sequential observation, while other lakes are observed sporadically and cited from Liu et al. (2021).

| LakeID | Lake | Observation date/period | Sources |
|---|---|---|---|
| TPL114 | Serling Co* | 18/04/2014~30/09/2014 | Guo et al. (2016) |
| TPL041 | Ngoring Lake* | 01/06/2011~29/11/2011 | Li et al. (2015) |
| TPL012 | Bangong Co* | 30/07/2012~21/10/2012 | Nation Tibetan Plateau Third Pole Environment Data Center |
| TPL028 | Dogze Co* | 19/08/2012~30/08/2013 | Nation Tibetan Plateau Third Pole Environment Data Center |
| TPL082 | Mapam Yumco | 25/09/2009, 04/09/2017 | Liu et al. (2021) |
| TPL031 | Tangra yumco | 06/09/2009 | |
| TPL101 | Pumoyong Co | 27/06/2009 | |
| TPL153 | Zhari Namco | 16/09/2009 | |
| TPL014 | Npen Co | 01/09/2009 | |
| TPL012 | Bangong Co | 24/07/2010 | |
| TPL069 | Langa Co | 14/07/2010, 08/09/2017 | |
| TPL048 | Kunggyu Co | 18/07/2010 | |

| | | |
|---|---|---|
| TPL098 | Paiku Co | 06/07/2010 |
| TPL121 | Taro Co | 23/09/2011 |
| TPL029 | Co Ngoin | 24/07/2012 |
| TPL149 | Yunbo Co | 12/08/2012 |
| TPL009 | Bam Co | 21/08/2012 |
| TPL027 | Dawa Co | 11/08/2012 |
| TPL154 | Zhangne Co | 17/08/2012, 18/10/2013 |
| TPL046 | Gomang Co | 17/08/2012 |
| TPL023 | Cuoe Lake/Co Ngoin1 | 02/08/2012, 24/06/2017 |
| TPL103 | Qagoi Co | 19/08/2012 |
| TPL028 | Dogze Co | 18/08/2012, 26/09/2013 |
| TPL070 | Lagkor Co | 08/08/2012 |
| TPL012 | Bangong Co | 28/07/2012 |
| TPL013 | Bandao Lake | 27/10/2012 |
| TPL003 | Amur Co | 22/10/2012 |
| TPL146 | Yongbo Lake | 03/10/2012 |
| TPL077 | Longwei Co | 25/10/2012 |
| TPL039 | Dogaicoring QangCo | 08/11/2012 |
| TPL008 | Ngangzi Co | 18/10/2013 |
| TPL055 | Gyado Lake | 26/10/2013 |
| TPL017 | Bura Co | 29/10/2013 |
| TPL030 | Tangqung Co | 23/09/2013 |
| TPL134 | Xuru Co | 02/09/2013 |
| TPL088 | Monco Bunnyi | 22/09/2013 |
| TPL114 | Serling Co | 07/08/2014 |

| TPL050 | Gozha Co | 25/09/2015 |
|--------|----------|------------|
| TPL038 | Dogai Coring | 07/11/2016 |
| TPL037 | Dorsoidong Co/Tu Co | 24/10/2016 |
| TPL122 | Nam Co | 24/06/2016 |
| TPL030 | Migriggyangzham Co | 29/10/2016 |
| TPL007 | Ngangla Ringco | 03/08/2017 |
| TPL067 | Gyaring Co | 02/07/2017 |
| TPL157 | Serling Co | 02/06/2017 |

[Figure]

Figure S1 Validation of modeling results against in-situ observation summarized in Table S1: (a)-(d) is the temporal validation respectively for the 4 lakes with sequential observation; (e) is the spatial validation for the 41 lakes with sporadic observation. The solid lines are the regression lines. The dashed lines are the 1:1 lines.

**References**

Guo, Y., Zhang, Y., Ma, N., Song, H., & Gao, H. (2016). NOTES AND CORRESPONDENCE: Quantifying Surface Energy Fluxes and Evaporation over a Significant Expanding Endorheic Lake in the Central Tibetan Plateau. Journal of the Meteorological Society of Japan, 94(5), 453-465.

Li, Z., Lyu, S., Ao, Y., Wen, L., Zhao, L., & Wang, S. (2015). Long-term energy flux and radiation balance observations over Lake Ngoring, Tibetan Plateau. Atmospheric Research, 155, 13-25.

Liu, C., Zhu, L., Wang, J., Ju, J., Ma, Q., Qiao, B., Wang, Y., Xu, T., Gao, H., Kou, Q., Zhang, R., & Kai, J. (2021). In-situ water quality investigation of the lakes on the Tibetan Plateau. Science Bulletin, 66(2021):1727-1730.

---

## Author Comment (AC3)

**An integrated dataset of daily lake surface water temperature over Tibetan Plateau**

Linan Guo[1,4], Hongxing Zheng[3], Yanhong Wu[1,2]*, Lanxin Fan[1,2], Mengxuan Wen[1], Junsheng Li[1,2], Fangfang Zhang[1], Liping Zhu[4], Bing Zhang[1,2]*

[1]Key Laboratory of Remote Sensing Science, Aerospace Information Research Institute, Chinese Academy of Sciences, Beijing 100094, China

[2]University of the Chinese Academy of Sciences, Beijing 100049, China

[3]CSIRO Land and Water, Canberra, ACT 2601, Australia

[4]Institute of Tibetan Plateau Research, Chinese Academy of Sciences, Beijing 100101, China

Table S1 Summary of *in-situ* lake surface temperature observation used for validation. The first 4 lakes denoted by * are with relative long sequential observation, while other lakes are observed sporadically and cited from Liu et al. (2021).

| LakeID | Lake | Observation date/period | Sources |
|---|---|---|---|
| TPL114 | Serling Co* | 18/04/2014~30/09/2014 | Guo et al. (2016) |
| TPL041 | Ngoring Lake* | 01/06/2011~29/11/2011 | Li et al. (2015) |
| TPL012 | Bangong Co* | 30/07/2012~21/10/2012 | Nation Tibetan Plateau Third Pole Environment Data Center |
| TPL028 | Dogze Co* | 19/08/2012~30/08/2013 | Nation Tibetan Plateau Third Pole Environment Data Center |
| TPL082 | Mapam Yumco | 25/09/2009, 04/09/2017 | Liu et al. (2021) |
| TPL031 | Tangra yumco | 06/09/2009 | |
| TPL101 | Pumoyong Co | 27/06/2009 | |
| TPL153 | Zhari Namco | 16/09/2009 | |
| TPL014 | Npen Co | 01/09/2009 | |
| TPL012 | Bangong Co | 24/07/2010 | |
| TPL069 | Langa Co | 14/07/2010, 08/09/2017 | |
| TPL048 | Kunggyu Co | 18/07/2010 | |

| TPL098 | Paiku Co | 06/07/2010 |
|---|---|---|
| TPL121 | Taro Co | 23/09/2011 |
| TPL029 | Co Ngoin | 24/07/2012 |
| TPL149 | Yunbo Co | 12/08/2012 |
| TPL009 | Bam Co | 21/08/2012 |
| TPL027 | Dawa Co | 11/08/2012 |
| TPL154 | Zhangne Co | 17/08/2012, 18/10/2013 |
| TPL046 | Gomang Co | 17/08/2012 |
| TPL023 | Cuoe Lake/Co Ngoin1 | 02/08/2012, 24/06/2017 |
| TPL103 | Qagoi Co | 19/08/2012 |
| TPL028 | Dogze Co | 18/08/2012, 26/09/2013 |
| TPL070 | Lagkor Co | 08/08/2012 |
| TPL012 | Bangong Co | 28/07/2012 |
| TPL013 | Bandao Lake | 27/10/2012 |
| TPL003 | Amur Co | 22/10/2012 |
| TPL146 | Yongbo Lake | 03/10/2012 |
| TPL077 | Longwei Co | 25/10/2012 |
| TPL039 | Dogaicoring QangCo | 08/11/2012 |
| TPL008 | Ngangzi Co | 18/10/2013 |
| TPL055 | Gyado Lake | 26/10/2013 |
| TPL017 | Bura Co | 29/10/2013 |
| TPL030 | Tangqung Co | 23/09/2013 |
| TPL134 | Xuru Co | 02/09/2013 |
| TPL088 | Monco Bunnyi | 22/09/2013 |
| TPL114 | Serling Co | 07/08/2014 |
| TPL050 | Gozha Co | 25/09/2015 |
| TPL038 | Dogai Coring | 07/11/2016 |

| | | |
|---|---|---|
| **TPL037** | Dorsoidong Co/Tu Co | 24/10/2016 |
| **TPL122** | Nam Co | 24/06/2016 |
| **TPL030** | Migriggyangzham Co | 29/10/2016 |
| **TPL007** | Ngangla Ringco | 03/08/2017 |
| **TPL067** | Gyaring Co | 02/07/2017 |
| **TPL157** | Serling Co | 02/06/2017 |

[Figure]

Figure S1 Validation of modeling results against in-situ observation summarized in Table S1: (a)-(d) is the temporal validation respectively for the 4 lakes with sequential observation; (e) is the spatial validation for the 41 lakes with sporadic observation. The solid lines are the regression lines. The dashed lines are the 1:1 lines.

[Figure]

Figure S2: Bias between reconstructed and MODIS-based LSWT in each month.

[Figure]

Figure S3: Time series of Dogze Co.

[Figure]

Figure S4: Long-term trends of summer (Jun-Aug), winter (Dec-Feb) and annual LSWT during 1978–2017.

**Table S2 Characteristics of satellite-based LSWT datasets for lakes across Tibetan Plateau**

| Data sources | Data | and | Period | Temporal | Number | Limitations | References |
|---|---|---|---|---|---|---|---|

| | methods | | resolution | of lakes | | |
|---|---|---|---|---|---|---|
| ARC-Lake | ATSR-2/AATSR-based | 1995-2012 | Daily | 112 | No data when temperature below 0℃ | Layden et al., 2015 |
| TPLakes_Temperature | MOD11A2-based lake-wide mean | 2000-2015 | 8-day | 374 | Shorter time span and lower temporal resolution | Wan et al., 2017 |
| Tplake_Temp | AVHRR-based with split-window approach | 1981-2015 | Daily | 97 | inconsistency due to revisit period and calibration among successive satellites | Liu et al., 2019 |

**References**

Guo, Y., Zhang, Y., Ma, N., Song, H., & Gao, H. (2016). NOTES AND CORRESPONDENCE: Quantifying Surface Energy Fluxes and Evaporation over a Significant Expanding Endorheic Lake in the Central Tibetan Plateau. Journal of the Meteorological Society of Japan, 94(5), 453-465.

Li, Z., Lyu, S., Ao, Y., Wen, L., Zhao, L., & Wang, S. (2015). Long-term energy flux and radiation balance observations over Lake Ngoring, Tibetan Plateau. Atmospheric Research, 155, 13-25.

Liu, C., Zhu, L., Wang, J., Ju, J., Ma, Q., Qiao, B., Wang, Y., Xu, T., Gao, H., Kou, Q., Zhang, R., & Kai, J. (2021). In-situ water quality investigation of the lakes on the Tibetan Plateau. Science Bulletin, 66(2021):1727-1730.

---

## Author Response (AR1)

Dear Editor:

Thank you very much for reviewing our manuscript titled "An integrated dataset of daily lake surface water temperature over Tibetan Plateau" (Manuscript ID: essd-2021-151). We have read carefully all the review comments and suggestions, which are valuable and very constructive for us to improve the manuscript. Point-by-point replies (in blue) to each of the comments are presented below.

We hope that our responses are sufficient and satisfactory.

Yours sincerely,

Dr. Yanhong Wu (wuyh@radi.ac.cn)
Dr. Bing Zhang (zb@radi.ac.cn)

**REPLY TO COMMENTS FROM THE REFEREES:**

**Referee: 1**

Guo et al. produced an integrated daily lake surface water temperature for 160 lakes across the Tibetan Plateau by combining satellite data and modeling and found that LSWT of lakes in the TP have increased significantly during the period 1978-2017. The result is interesting and important to understand the response of lakes to climate changes. Some major comments are as the following:

1: validation of the modeling result is important at some specific lakes. For example, in situ observation data are available at Dazeg Co, Bangong Lake, Nam Co, Paiku Co etc. Why not use the in situ data at these lakes to compare with satellite or modeling result. It is also very useful to compare the modeling result with satellite data at specific lakes, rather than just like that in figure 4.

Reply: Thank you for the comments. We strongly agree that it is valuable to validate the modelling results against the in-situ lake surface temperature observation in the Tibetan Plateau. The in-situ observation data however are not widely available for lakes in the Tibetan Plateau to conduct an overall validation of the modelling results. Nevertheless, as suggested by the reviewer, we have compared the modelling results against the currently publicly available in-situ surface water temperature for lakes in the Tibetan Plateau, which include sequential observation of 4 lakes (i.e., the Ngoring Lake, Serling Co, Dogze Co, Bangong Co) and sporadic observation of 41 lakes (Table S1 in Suplementary). As shown in Fig. 6, the simulated lake surface temperature is in good agreement temporally with the sequential observations ($R_2$=0.97, 0.92, 0.90, 0.97 for Ngoring Lake, Serling Co, Dogze Co, Bangong Co respectively) and spatially with the sporadic observation (R2=0.94). For the validation against the sporadic observation, simulated lake surface water temperature of day same to the observation is used. We would like to clarify that Figure 4 is presented to show the consistency of the two satellite-based observations for specific lakes. The comparisons between satellite-based observation and simulation for each specific lakes are summarized at Figure 5.

2: the method of satellite data (part 3.1) is not clear enough for me. As we know that, lakes in the TP are often coved by cloud and MODIS LSWT is easily affected, especially in summer. From my experience, the quality of MODIS LSWT is low for many lakes in summer due to the cloudy weather and anomalous value of LSWT can be produced. The authors do not mention how they deal with this. When there is missing data, how do you calculate seasonal or annual LSWT and their trend?

Reply: Thank you for the comments. We noticed that there are missing data in the MODIS LST, and the quality of the MODIS LST are not always of good quality due to issues mentioned by the reviewer. The limitation of the MODIS LST data (i.e., missing data, quality, and of relative short period) necessitate the research to reconstruct the long-term LST via model simulation to investigate the temporal and spatial variation of LST across the Tibetan Plateau. In this research, therefore, the seasonal and annual LSWT as well as their trends are assessed not based on the MODIS LST but based on the reconstructed LST. The MODIS LST is used solely to calibrate and validate the simulation, which makes the simulation possible and reliable.

3: About the quality of reconstructed LSWT by air2water. As the author mentioned, the daily air temperature were interpolated from the nearest meteorological stations with elevation adjustment. In fact, there are very few meteorological stations in the vast west part of TP, while most lakes are located here. So the reliability of the modeling result need to be further tested.

Reply: Thank you for the good question. We deeply agree that the spread of the meteorological stations in the Tibetan Plateau is a big constraint for earth system researches in the region. We noticed that quite a few researches efforts have been invested to provide reliable reanalysis climate datasets for the entire TP (e.g., China meteorological forcing dataset, DOI: https://doi.org/10.11888/AtmosphericPhysics.tpe.249369.file). In this research, however, we used the best available ground-based climate observations together with a simple interpolation approach (the nearest neighbor with elevation adjustment) to provide model inputs for LSWT simulation across the TP. This is because that the public available climate reanalysis datasets were found not necessarily suitable for the LSWT simulation when we compared them against the observations from the meteorological stations. The results however were not demonstrated in our manuscript as it is out of the scope of the study. In addition, in the future research, it is worth exploring the effects of spatial interpolation algorithms for climate variables on the performance of LSWT modelling across the TP with specific considerations on the number of donor stations and the spatial distance between the lakes and the donor stations. We believe that it would be an interesting method-oriented research topic, from which the results could contribute to improve the understanding the spatial variability of climate in the Tibetan Plateau.

**Referee: 2**

The authors present a dataset with lake temperature data for the Tibetan Plateau (TP). They extrapolated lake surface water temperatures (LSWT) as measured by satellite sensors back in time to obtain a complete LSWT dataset also for the time prior to 2000 when such data became available from the MODIS sensor. They used a simple physical model for this purpose which was trained with MODIS LSWT and forced by air temperature measurements from the extremely sparse station network on the TP.

Both the data itself and the data article would benefit greatly from a more detailed analysis and description of uncertainties, model limitations, resulting bias/patterns etc. to prove that this data is reliable and may be used by others for the varied purposes mentioned in the paper introduction.

Reply: Thank you very much for your in-depth review for our manuscript. We have revised the manuscript carefully according to all your comments and suggestions.

**There are three major aspects I strongly recommend the authors to address:**

**1)** Data presentation: This paper is about time series of temperature data, but it does not provide any time series plots, only maps where each time series is reduced to a single point. The maps are useful to get a spatial overview and reveal spatial patterns, but they are not at all suitable to assess/present any temporal or seasonal aspects and also simply don't show how the data looks like. This makes it impossible for the reader to judge whether this data would suit their purpose without downloading and analysing the data themselves - which ought to be the main purpose of a data article.

Reply: Out dataset include daily temperature time series for 160 lakes across the Tibetan Plateau, which is not very suitable to plot lake by lake in the manuscript. As summaries of the dataset, we have shown in Figure 3, Figure 5 and Figure 7 for the performance of the MODIS-based LSWT and the reconstructed LSWT by evaluating the data over the temporal dimension (as indicated by NSE and bias) and its long-term trend, which is straightforward for the readers to assess the quality of the data and understand the temporal variation of LSWT in the TP. We agree that, however, the performance metrics (NSE or bias) and the trend may not reflect other aspects like seasonal variation of the LSWT. For such a case, as suggested, we have shown in this Supplementary time series plot of the Dogze Co (Figure S5), which is the lake where in-situ observation is available for the period 2012-2013. In Figure S5, the temporal variations of LSWT at both daily and annual scales are shown, and the reconstructed data is compared against the in-situ observation as well. We also include time series plot for each lake in our updated dataset for readers to have a quick view (https://doi.org/10.5281/zenodo.5878436).

I suggest additional data quality checks/analyses/comparisons over time, for different seasons and temperature ranges. It would be especially useful to compare the modelled data with independent measurements/data for the time span 1978-2000.

Reply: Thank you for the comments. Summaries of model performance evaluation for each month are now presented in Figure S4 in the Supplementary. We agree that it is valuable to check the modelling results when/where the in-situ observations are available. The in-situ observation data however are not widely available for lakes in the Tibetan Plateau to conduct an overall validation of the modelling results. Nevertheless, we have compared the modelling results against the currently publicly available in-situ surface water temperature for lakes in the Tibetan Plateau, which include sequential observation of 4 lakes (i.e., the Ngoring Lake, Serling Co, Dogze Co, Bangong Co) and sporadic observation of 41 lakes (Table S1). As shown in Fig. 6, the simulated lake surface temperature is in good agreement temporally with the sequential observations ($R^2$=0.97, 0.92, 0.90, 0.97 for Ngoring Lake, Serling Co, Dogze Co, Bangong Co respectively) and spatially with the sporadic observation ($R^2$=0.94). For the validation against the sporadic observation, simulated lake surface water temperature of the date same to the observation is used.

The data itself: A brief analysis of the T time series in the TPLxxx.csv datasets revealed the following points, all of which should be addressed in the article in order to properly characterise the data:

- for most lakes, the model data does not differ much between different years, and for many lakes it even seems to consist of a series of nearly exact clones of annual T curves. These often have a characteristic kink in spring which deserves to be adressed to exclude bias/inappropriate model behaviour (these kinks are at ca. -2, so likely related to thawing processes, but then again this feature doesn't show up on all lakes and seemingly also not in the MODIS LSWT data).

Reply: As summarized and shown in Fig.7, for the period 1978–2017, most of the lakes (except for Lake Beidao and Lake Changhong) showed significant increasing trends in annual LSWT with increasing rate ranging at 0.01 to 0.47 ℃ 10a$^{-1}$. In general, lakes in the south are found to have relative higher warming rate than those in the north of the Tibetan Plateau. The results though indicate consistent warming trends for lakes across the TP, as also noticed by the reviewer, the seasonality (i.e., intra-annual variation) of the LSWT doesn't show significant changes in the studied period, which is because the seasonality is

largely determined by the relative constant seasonality of solar radiation. We agree that the kink in spring for some lakes could be related to the thawing processes. It is also because that the lakes may take some time to warm up in the spring depending on their heat capacities, which are in association with the area and volumes of the lakes. For the MODIS LSWT data, since there are quite a few missing data in it due to the limitation of the sensors, the kink cannot always be observed.

- the modelled data are way more smooth in time than the MODIS LSWT data and don't reproduce any extreme values or variability.

Reply: As noticed also by other researchers (Zhang et al., 2014; Wan et al., 2017), we'd like to point out that the MODIS LSWT data is often with more extreme values and larger variability particularly when compared against the in-situ observations (e.g., Figure S5). The MODIS daily data inherently include outliers due to the reliability of the sensors and effects of clouds and solar altitude, which could result at overestimation of the daily variability in LSWT. It should also be noticed that the MODIS-based observation is a measure of instant LSWT (i.e., LSWT of the specific time in a day when satellite pass over the lakes) instead of daily mean of the LSWT. It therefore could be peakier than the modelled daily mean LSWT. It is important to note that satellite measures only water temperature of the top surface ($\sim$10– 20µm deep) temperature, known as skin temperature. The skin temperatures differ from surface water temperature because the thermal structure of the water column but are relevant and sufficient to demonstrate spatial-temporal patterns of surface water temperature for lakes and conducive to model calibration and validation (e.g., Andréassian et al., 2012; Prats and Danis, 2017; Prats et al., 2018).

- for the period before 2000, there seems to be a fixed annual maximum temperature for each lake that doesn't change.

Reply: That is not quite true. For confirmation, we have checked the dataset by calculating the trend of the annual maximum LSWT of all lakes from 1978 to 1999 (period before 2000). The results show that the maximum LSWT of lakes changed substantially during the period, where 150 lakes show significant warming trends. The warming rates of maximum LSWT are between -0.1~0.62℃ 10a$^{-1}$ across the region. The results are consistent with the warming trends represented by their corresponding maximum air temperature, where the increasing rates of maximum annual air temperature are between -0.3~1.33℃ 10a$^{-1}$ across the TP. The variation rates of the maximum water temperature are lower than that of the maximum air temperature thanks to the heat storage capacity of the lakes.

- there are gaps in the MODIS LSWT data, but it is nowhere described where these come from.

Reply: The gaps (which marked as "-999" in the dataset) comes from MOD11A1 product itself. This daily product has many missing values because of cloud effects and the limitation of the algorithm (Wan et al., 2013). In this revision, we have described the issues with more details (see Line114).

- No data/temperature uncertainties are provided. An uncertainty range/measure should be included at least for the model data.

Reply: We recognize that uncertainties may exist in the reconstructed LSWT due to uncertainties in model inputs (i.e., air temperature) and model parameters. However, we prefer to provide the currently "best" possible LSWT dataset for the lakes across the TP. This is largely because that there are few in-situ observations available to determine a rational uncertainty range for the LSWT. We however agree it is valuable to quantify the uncertainty of the modelled LSWT when/where the in-situ observations becoming more available.

**2)** LSWT: To give the readers an idea on the nature and reliability of this data, the introduction should contain an explanation of LSWT - what it is, how it is measured, including a description of the different available satellite sensors/datasets (with special emphasis on the data and validation products used in this study), and what the limitations of such measurements are.

Reply: Thanks for the comments. As suggested, we have included in this revision the meaning of LSWT and more details about the available satellite dataset (see Section 3.2 Line 155-160 and Line 41-53).

Also, you mention several other LSWT datasets (Wan et al., 2017, Zhang et al., 2014) and that yours is different - but different in what regard, and how does it compare to these datasets? I am missing a clear introduction on the differences and strengths/weaknesses/advantages of the existing datasets to justify this new dataset, and most of all to help users decide whether this is the right data for their purpose.

Reply: The uniqueness of our dataset herein is that it combines both the MODIS-based and model-based LSWT. To our knowledge, our model-based daily LSWT is currently the first effort in providing region-wide daily LSWT dataset for lakes across the Tibetan Plateau. The dataset is a complete dataset covering a long temporal span (1978-2017). The existing datasets in the literature are largely based on remotely sensed data, which may cover a shorter time period, have lower temporal resolution and lots of missing data (as summarized in Table S2 in the Supplementary).

**3)** Model: More background on the air2water model would help to prove that this model is fit for the purpose and can produce reliable extrapolation results. This also includes the forcing data: are the sparse station measurements really fit for the purpose? From the information given right now, it cannot be judged whether this model was used as intended by the original authors - or taken out of context/pushed beyond its limitations.

Reply: The Tibetan Plateau is a region with limited ground observation and challenging for more complex modelling which requires more data to configure, calibrate and validate the models. This is particularly difficult when the purpose is to develop a dataset for most of the large lakes in the TP. The air2water model developed by Piccolroaz et al. (2013) is a hybrid model with strong physical base but simplifies the thermodynamic equations to minimize the input requirement while preserving the robustness of deterministic models. air2water has been shown to provide similar performance, in terms of simulating LSWT, to process-based models , and to be an effective tool to investigate LSWT responses to historic and future climate change (Piccolroaz et al. 2015; Piccolroaz et al. 2016; Schmid and Köster 2016;Wood et al. 2016; Czernecki and Ptak 2018; Piccolroaz et al. 2018; Piccolroaz and Toffolon 2018) also when applied to lakes with different morphological characteristics from around the world (Toffolon et al. 2014; Prats and Danis 2019). As suggested, more background on the air2water model has been added in this revision (see Line 62-67).

Piccolroaz S, Toffolon M (2018) *The fate of Lake Baikal: how climate change may alter deep ventilation in the largest lake on Earth. Clim Chang 150:181–194.*

Piccolroaz S, Healey NC, Lenters JD et al (2018) *On the predictability of lake surface temperature using air temperature in a changing climate: a case study for Lake Tahoe (U.S.A.). Limnol Oceanogr 63:243–261.*

Toffolon M, Piccolroaz S (2015) *A hybrid model for river water temperature as a function of air temperature and discharge. Environ Res Lett 10:114011.*

Wood TM, Wherry SA, Piccolroaz S, Girdner SF (2016) *Simulation of deep ventilation in Crater Lake, Oregon, 1951–2099. US Geol Surv Sci Investig Rep 2016–5046:43.*

**Detailed comments (P=page, L=line)**
* * *
Data:
* * *
- lake_info.csv: Please add the encoding of the .csv file (to the readme, for example) - common international encodings corrupt the Chinese signs which may cause trouble for users

Reply: The "lake_info.csv" file has been updated by removing the Chinese characters.

Abstract
* * *
- data for 160 lakes from RS and modelling: what RS data, what (kind of) model? Please state here.

Reply:Added. The model refers to air2water and RS data is the MODIS data.

- validated against in-situ observations for 2000-2017, extended back to 1978

Reply: This means that we calibrated and validated the model for the period 2000-2017 where satellite observations are available. We then extended the LSWT dataset based on the calibrated model.

- comparable to LSWT from AVHRR and ARC-Lake - has there been done a comparison, or you just claim that? If yes: give some numbers, if not: please do a comparison.

Reply: The comparison is done and the results (represented by NSE) is presented in Section 4.1.

- "unique for its time span and resolution" - what is the resolution? Please state here.

Reply: It means temporal resolution.

- trend analysis: warming rate increases from 0.01 to 0.4 per decade - for which periods? Please clarify here.

Reply: We have mentioned the period is 1978-2017 in Page1 Line 21.

- mostly in winter - what does "more evident" mean? Please clarify here.

Reply: Corrected as "The warming rate is higher in winter than in summer".

- "integrated data and methods" - what methods? Please clarify already in the abstract.

Reply: The methods mean using remotely sensed data to calibrate/validate the model, and then use the model to reconstruct the daily LSWT time series.

Introduction

- What is lake surface water temperature, in this context? Please explain what the satellites measure, and what in-situ sensors measure (since your dataset is based on both). This is important background to understand the data. When comparing/validating with in-situ measurements, as in this paper, you need to consider that in-situ- and satellite-based sensors usually measure something different, and address this.

Reply: As suggested, we have included in this revision the meaning of LSWT and more details about the available satellite dataset (see Section 3.2 Line 155-160 and Line 41-53). The in-situ surface temperature at depth z (Tz), also known as (bulk) surface temperature, is the temperature measured at the said depth.

- p2, L38: but some TP lakes have in-situ data? Please add some more information on that, as this would provide a very important valildation data source for this paper. Have you consulted this data? If not, why not?

Reply: We have collected some in-situ observations from the literature to validate our dataset. Summary of the in-situ observations are shown in Supplementary Table S1, and the validation results are shown in Figure 6.

- P2, L41: a reference from 2010 looks a bit outdated for a claim on "increasingly higher temporal and spatial resolutions" in 2021 as a lot has happened since that paper was written. Reconsider its relevance / correct placing in the text.

Reply: References corrected and added.

- P2, L141ff: What I am missing here is an overview over sensors that measure LSWT. Both the sensors/data sources in themselves (i.e. individual data/scenes), and for the datasets you mention here.

Reply: There are multiple approaches to estimate land/water surface temperature from the brightness temperatures (BT). In the case of inland water bodies like large lakes, the most common approach is the

split-window technique where the difference between the two adjacent thermal channels (10.5–11.5 µm, 11.5–12.5 µm) is taken as a measure of atmospheric attenuation to derive the Surface Temperature (ST). The accuracy of the derived ST depends on the split-window coefficients, which in turn is based upon multiple parameters like spectral response function and emissivity of the channels, column water vapour in the atmosphere and View Zenith Angle (VZA) of the sensor. These coefficients are derived by regressing simulated BT's from radiative transfer models like MODTRAN against the atmospheric profiles. The coefficients are generally derived on a regional or global level for the estimation of Land Surface Temperature (LST) and Sea Surface Temperature (SST). For lakes, both LST and SST based algorithms are used interchangeably and with higher accuracy of RMSE between 0.5–1.5 °C. Furthermore, lake and sensor specific constants are published for an exhaustive list of global lakes (Hulley et al., 2011).

- P2, L45f: How can these datasets be global and nevertheless spatially incomplete on the TP? Please clarify.

Reply: The datasets are global (as claimed by their authors) in terms that they cover most regions in the world. It however doesn't mean they cover every corner of the earth particularly for regions like Tibetan Plateau, where the necessary data or model may not sufficient for the researchers during their work.

- P2, L47: What sensors?

Reply: The information about the sensors is added at Line 49.

- P2, L61ff: OK, you go for the simplified model becuase you have barely any in-situ observations and this model still works with limited in-situ data - but what input is your model based on, then? This doesn't sound so convincing. Please rewrite with examples or some more information to clarify (for what area/setup has this model been tested/developed, for the TP or somewhere else?).

Reply: In this study, the model is driven by air temperature and calibrated against remotely sensed LSWT. The simulation results were validated by both in-situ observation (for lakes that in-situ data could be collected) and MODIS-based lake surface water temperature (for all lakes). The Tibetan Plateau is a region with limited ground observation challenging for more complex modelling which requires more data to configure, calibrate and validate the models, particularly when the purpose is to develop a dataset for most of the large lakes in the TP. The air2water model developed by Piccolroaz et al. (2013) is a hybrid model (Toffolon and Piccolroaz 2015) with strong physical base but simplifies the thermodynamic equations to minimize the input requirement while preserving the robustness of deterministic models. The air2water has been shown to provide similar performance, in terms of simulating LSWT, to process-based models , and to be an effective tool to investigate LSWT responses to historic and future climate change (Piccolroaz et al. 2015; Piccolroaz et al. 2016; Schmid and Köster 2016;Wood et al. 2016; Czernecki and Ptak 2018; Piccolroaz et al. 2018; Piccolroaz and Toffolon 2018) also when applied to lakes with different morphological characteristics from around the world (Toffolon

et al. 2014; Prats and Danis 2019). As suggested, more background on the air2water model has been added in this revision (see Line 62-67).

Piccolroaz S, Toffolon M (2018) *The fate of Lake Baikal: how climate change may alter deep ventilation in the largest lake on Earth. Clim Chang 150:181–194.*

Piccolroaz S, Healey NC, Lenters JD et al (2018) *On the predictability of lake surface temperature using air temperature in a changing climate: a case study for Lake Tahoe (U.S.A.). Limnol Oceanogr 63:243–261.*

Toffolon M, Piccolroaz S (2015) *A hybrid model for river water temperature as a function of air temperature and discharge. Environ Res Lett 10:114011.*

Wood TM, Wherry SA, Piccolroaz S, Girdner SF (2016) *Simulation of deep ventilation in Crater Lake, Oregon, 1951–2099. US Geol Surv Sci Investig Rep 2016–5046:43.*

- P2, L66: LSWT from which sensor? and does the air2water model work with LSWT data, if it was developed for / is aiming at in-situ measurements?

Reply: The only input to the air2water model is air temperature. The remotely sensed LSWT is used to calibrate and validate the model.

Study area

- P3, L75: why "most" but not all, how did you select?

Reply: The lakes are selected according to the data availability, either in-situ air temperature data or the MODIS data. Lakes without altitude information or efficient remote sensing data (data points account for less than 40% during the period) for model calibration are neglected.

Methods
* * *
- P4, L90ff: What is the accuracy/uncertainty of this data? (How much can one trust these temperatures? Any danger for a consistent bias?)

Reply: We recognize that uncertainties may exist in the reconstructed LSWT due to uncertainties in model inputs (i.e., air temperature) and model parameters. However, we prefer to provide the currently possible "best" LSWT dataset for the lakes across the TP. This is largely because that there are few in-situ observations available to determine a rational range for the LSWT. We however agree it is valuable to quantify the uncertainty of the modelled LSWT when/where the in-situ observations becoming more available. For the MODIS-based LSWT, it has been reported in other researches that the accuracy is better than 1 K (Coll et al., 2005; Wan et al., 2002). For some lakes in the Tibetan Plateau, the biases of the MODIS-based LSWT were reported to be around −1.74 °C (Song et al., 2016) or −1.4 °C (Zhang et

al., 2014), which are consistent with our research herein. The description has been added in Line 98-101.

Coll, C., et al. Ground measurements for the validation of land surface temperatures derived from AATSR and MODIS data, Remote Sens. Environ., 97(3), 288–300 (2005).

Wan, Z., Y. Zhang, Q. Zhang, and Z.-L. Li, Validation of the land-surface temperature products retrieved from Terra Moderate Resolution Imaging Spectroradiometer data, Remote Sens. Environ., 83(1–2), 163–180 (2002).

- P4, L93: what do the times in brackets mean? Night equator overpasses? Please clarify.

Reply: MODIS (Moderate-resolution imaging spectroradiometer) aboard the Terra and Aqua satellite. The satellite Terra overpass twice each day at 10:30 AM and 10:30 PM, while the Aqua overpass at 1:30 AM and 1:30 PM. Here we use the product from MODIS aboard Terra.

- P4, L95f: what is the difference between MOD11A2 and -A1? Only A1 is explained.

Reply: The MOD11A2 is an 8-day product, which is the averaged LSTs of daily MOD11A1 products over 8 days and includes up to 46 data points per year.

- P5, L100f: different in what regard? Just the method, or also the temperatures you get - how do they compare, and why is your method better? Please clarify.

Reply: Zhang et al. (2014) treated the temperature of centroid pixels as LSWT, but only one pixel could not represent the whole lake. In our research herein, the lake-wide mean temperature is calculated and considered to be more reliable.

- P5, L106: Are there data gaps, and/or was there any filtering applied to the data, both in space and time?

Reply: There is no gaps in the reconstructed lake surface temperature as the input air temperature is with few gaps and has been filled by linear interpolation before running the air2water model. Details was shown in Line 148.

- P5, L110: How are these surface temperatures measured? How accurate/reliable/representative are the measurements? This plays a huge role for your model setup and thus dataset, as very few stations are forcing the model for the majority of the lakes. And how long are the records?

Reply: We used the ground-based observation of air temperature for the period 1978-2017 to drive the model.

- P5, L125: As far as I am aware, many of the TP lakes are very shallow. Is a deep water temperature of 4 degrees realistic at all times of the year?

Reply: In the air2water model, the deep water temperature ($T_h$) is assumed to be a constant rather than a time-varying parameter, which is because that the model performance is largely insensitive to $T_h$ and it is practically difficult to measure $T_h$ of the studied lakes. In principle, the effects of $T_h$ on model performance can be complemented by model parameters like the maximum thickness of the mixed layer ($D_r$) in the model, which can be calibrated to account for the thermodynamics in the lake (Toffolon et al., 2014).

- P5f, L126ff: please explain briefly in words what this model does, and what parameters play the strongest role here, for those users who are not that familiar to read/think in equations.

Reply: The air2water is a semi-physical model designed to simulate lake surface temperature principally based on lake surface heat balance. The model is driven only by air temperature and preferably also depth of lake. For the original model (Piccolroaz et al., 2013), $a_{1-4}$ in Eq. (2) are the most sensitive parameters determining model performance. As suggested, we have explained this at Line 136.

- P6, L138f: there are very few stations on the TP, and none whatsoever in 1/4 of the area you look at, and where the majority of the adressed lakes lie. At the same time, the TP has a very varied topography and greatly varying influence of different weather systems (I/E Monsoon, Westerlies). Interpolation between "nearby" stations doesn't seem valid at all to me, questioning the entire model setup and thus the entire dataset. Have you considered uncertainty in these data? How much do the modelled lake T change if you alter the interpolated air T input (sensitivity analysis)?

Reanalysis data are maybe not perfect either in this area (due to too little measurement input), but they consider way more physics in deriving the temperature spatial pattern than a simple spatial/elevational interpolation. Have you tried to downscale air T for the lake locations from reanalysis data? How does that change the modelled lake T?

Reply: We noticed that quite a few researches efforts have been invested to provide reliable reanalysis climate datasets for the entire TP (e.g., China meteorological forcing dataset, DOI: https://doi.org/10.11888/AtmosphericPhysics.tpe.249369.file). In this research, however, we used the best available ground-based climate observations together with a simple interpolation approach (the nearest neighbor with elevation adjustment) to provide model inputs for LSWT simulation across the TP. This is because that the public available climate reanalysis datasets are found poorer for reconstructing the LSWT as compared against the simulations based on the observed (or interpolated) air temperature from meteorological stations (the results are not demonstrated in the manuscript as it is out of the scope of the study). Though it is worth exploring the effects of spatial interpolation in providing reliable air temperature for LSWT modelling, the improvement in the model performance and in accuracy of the dataset could be marginal because of the limited ground-based observations in the Tibetan Plateau. It should be noticed that, however, in this research we are aiming at providing reliable LSWT rather than air temperature for lakes in the Tibetan Plateau, where the quality of the dataset have been validated by comparing to both the satellite-based and the in-situ observations.

- P6, L145f: Please introduce that dataset (and AVHRR) properly so that the readers can judge your comparison/validation.

Reply: In this Methods section we focus more on introducing the methods. More details about the dataset itself could be found in the following sections "Results and discussions" and "Data availability". Moreover, as suggested, we have added more details about the other datasets in Table S2 in Supplementary.

Results and discussion
* * *
P6, L151: The ARC dataset also needs to be introduced properly in order for the readers to be able to judge your comparison/validation. What about in-situ lake T measurements (which you mention in the introduction)? These would provide a really valuable validation dataset.

Reply: Description about the ARC-Lake dataset has been added in Line181-185. Details about the in-situ data has been introduced in Line 214-218.

P7, L156f: An R2 of 0.6 isn't great, and "highly comparable" doesn't sound convincing when the temperatures should be identical if both datasets were correct - and both datasets are from remote sensing (not completely different ways of measuring), so one wouldn't expect such big differences.

Reply: We have reworded the statement here as "that the LSWT from the two sensors are largely comparable with each other for the studied lakes". As can be seen in Figure 3, for most of the lakes (80%), the NSE is higher than 0.7, which suggests statistically significant correlations between the datasets.

P7, L157: How did you compute the bias, and which of the two datasets is warmer? Consistent biases of -3 to +5.6 degrees seem very much.

- P7, Fig. 3b: all except one station have a positive bias, for many lakes, the bias is <2.5 degrees. This seems a lot. Why could that be?

Reply: The bias was the mean value of the difference between MODIS-based LSWT and AVHRR-based LSWT, which means the MODIS-based result is warmer. The reason for the relative high bias should be that "TPlake_Temp" shows an abrupt shift at 1995, which was assumed caused by the inconsistency calibration among successive satellites (Liu et al., 2019) thus affect the reliability of the dataset. The author (Liu et al., 2019) of this dataset also suggested that "the data cannot be applicable for climate change studies without ensuring that the calibration of the TP LSWTs from successive satellites is consistent, using overlap periods of the satellites."

- P7, Fig. 3a: Some of the poorest agreement between your data and TPlake_Temp data are clustered on the NW TP. Why, have you looked into that?

Reply: The three lakes with relative lower NSEs found in the NW TP are Margai Caka, Jianshui Lake and Yang Lake, where the AVHRR-based LSWT are with much larger data gaps, particularly with little records for the winter season. Hence, the NSEs of the three lakes largely reflect model performance in the warmer seasons, which could be poorer than the overall model performance.

- P7, L166ff: not sure whether I understand correctly: these 11 plots include all lakes within the ARC dataset within your study area? Which lakes are these? Adding a map as in Fig. 3 for the comparison with the ARC dataset would help greatly for the readers to compare the data more easily.

Reply: Lakes with both daytime and night-time LSWT in ACR-Lake are comparable with our LSWT from MODIS, thus only 11 lakes are available within our study area. We added the location of the 11 lakes in Figure S1 (in Supplementary).

- P8, Fig. 4: The datasets agree well in a linear way up to a point at ca. 10-15 deg C ARC-Lake T, where the relationship suddenly changes. Why is that?

Reply: We have found that the LSWT time series from ARC-Lake are much smoother than the MODIS-based one and the maximum annual LSWT tends to be constant. Meanwhile, the MODIS-based LSWT is peakier and with more outliers. We think this is the unknown limitation of the satellite-based observations and their corresponding algorithms that need to be improved.

- P8, Fig. 4: How did you compute the bias here? As the fitting line is not 1:1, the bias increases/decreases with measured T, which should be addressed.

Reply: The bias was the mean value of the difference between MODIS-based LSWT and ARC-Lake LSWT. The interceptions of the fitting line suggest systematic bias of the two sensors.

- P8, L176f: please state these numbers for the validation period, excluding the training period.

Reply: Thanks. We have confirmed that these numbers are for the validation period (excluding the training period) in Line 197. The bias for training (or calibration) is shown in Figure S2 in Supplementary.

- P8, L177ff: again, what do you mean by bias? And how do the results compare for different times of the year, or different water temperatures? Or distance to the weather stations (forcing data)? These aspects reflect model parameters better than latitude/elevation etc.

Reply: The bias herein refers to overall bias for the period 2000-2017, where we conduct pair-wise comparison between the simulated LSWT and the MODIS-based LSWT using all the available data pairs without considering model performance for different seasons or different magnitude groups of temperature. We appreciate the idea to investigate differences of model performance in accordance to time or magnitude of temperature. Summaries of model performance evaluation for each month are now presented in Figure S4 in the Supplementary. And we have added in the manuscript (Lines 204-205) that the simulated dataset shows more reliable in July and Nov, where the median bias is the smallest.

- P9: what is the uncertainty of your temperature data?

Reply: The uncertainty of the temperature data is currently challenging to quantify mainly due to the lack of in-situ observations in the Tibetan Plateau. Nevertheless, the temperature data has been assessed by comparing against the MODIS-based LSWT (Figure 5) and the publicly available in-situ observations (Figure 6). It should be noted that, the MODIS-based (or other satellite-based) LSWT is a substitute of the in-situ observation but, by definition, it is not complete the same to the modelled one. The satellite-based LSWT usually refers to the skin temperature of lakes for a specific time and that modelled is mean daily LSWT.

- P9: it is hard to get a feeling of the data from these maps only. I am missing time series from a few example lakes with different location/elevation/temperature/size/performance.

Reply: Example time series of Dogze Co are presented in Figure S5 in the Supplementary, showing daily and annually variation of the reconstructed LSWT and its comparison against the MODIS-based LSWT and in-situ observation as well. We also include time series plot for each lake in our updated dataset for readers to have a quick view (https://doi.org/10.5281/zenodo.5878436).

- P9, L197: label these lakes in the map/figure (the same accounts for other lakes mentioned in the text)

Reply: Done.

- P10, L198ff: also here, some time series plots would be useful to show these trends. Are they visible in the entire data range, or just part of the data (for example the period where you have measurements)? How do they compare to the forcing air T trends, are the trends shown here maybe simply reproducing air T trends? This aspect deserves discussion to undermine the validity and reliability of your trend analysis.

Reply: See reply above and Figure S5 in the Supplementary. Time series plots for each lake have been attached to the updated dataset (https://doi.org/10.5281/zenodo.5878436). As described above, the warming rates of annual mean LSWT are between 0.01~0.47℃ 10a$^{-1}$ across the region, while the increasing rates of annual mean air temperature are between 0.03~0.09℃ 10a$^{-1}$ across the TP. The warming trends of lake are consistent with air temperature but with smaller increasing rates implicating the impacts of heat storage capacity of the lakes. We added this information in Line 228-231.

- P10, L202ff: you argue that cold glacier water could have led to decreasing summer water T but state no source for this - and as a matter of fact, the glaciers in this part of the TP have been growing in recent decades. I thus suggest you remove speculations on the reason of the pattern (alternatively, you need to considerably extend this discussion and undermine it with literature).

Reply: Thanks for the comments. We have removed the statement from there.

Data availability
* * *
- P11, L216: Nice that you are sharing your code! A quick recap on what this contains (data preprocessing only, none of the modelling?) could be useful here. I quickly checked the code and see there's a variable error on line 14, you may want to fix that.

Reply: Here we share the GEE code that extracted the MODIS-based LSWT which is the calibration data of the modified air2water model. The original air2water is open source at https://github.com/spiccolroaz/air2water. We will also wrap our modified model codes and share them at GitHub, the link to which will be added to the final revision of the manuscript. We double checked the GEE code and didn't think there is a variable error.

Conclusions
* * *
P11, L226: "successfully": You never show a validation for the period 1978-2000, thus I disagree with that claim.

Reply: There is little in-situ LSWT observation in TP available for validation against the period 1978-2000. Nevertheless, this limitation doesn't stop us "successfully" reconstructing the LSWT for that period as the reconstructed dataset has been validated against the in-situ observation for the period 2009-2017 and against the MODIS-based LSWT for the period 2013 to 2017. For more precise expression, we have reworded the statement as "The modified air2water model is found successfully in reconstructing the daily LSWT of lakes across the Tibetan Plateau by extending the LSWT time series to the period 1978–2017 and filling the time gaps in the satellite-based LSWT though with uncertainties from model inputs and model parameterization." (Line 261-263)

Figures
* * *
All figures need to clearly state what data period they represent (not currently the case). Also, in some cases the used colour maps are rather little intuitive (usually, green = good and red = bad). Consider a single colour / two-colour diverging scale, varying symbols or symbol sizes to improve readability.

Reply: All figures have been remade.

---

## Author Response (AR2)

Dear Editor:

Thank you very much for reviewing our manuscript titled "An integrated dataset of daily lake surface water temperature over Tibetan Plateau" (Manuscript ID: essd-2021-151). According to your suggestions, we have made minor revision in this version by addressing the comments in referee report #1. The reply (in blue) to the reviewer's comments is presented below. We hope that our revisions and responses are sufficient and satisfactory.

Yours sincerely,

Dr. Yanhong Wu (wuyh@radi.ac.cn)
Dr. Bing Zhang (zb@radi.ac.cn)

**REPLY TO COMMENTS FROM THE REFEREES:**

The revision has been greatly improved. However, my concerns are still not fully addressed. The paper lacks the comparison of the modeling result with in-situ observation. Correlation analysis has been done, but it is not sufficient for me to evaluate the quality this data. A Figure about comparison of modeling result with in-situ observation should be added. As far as I know, there are several lakes with in-situ water temperature observation. If there is no figure about this, the read can not judge the quality of the data.

Reply: The in-situ water temperature observation data are not widely available for lakes in the Tibetan Plateau. Nevertheless, to ensure high quality of our dataset, we have compared the dataset against the best publicly available in-situ water temperature observations in the Tibetan Plateau, which include sequential observation of 4 lakes and sporadic observation of 41 lakes covering the period from 2009 to 2017 (see L214-219 in the manuscript and Table S1 in the Supplementary).

As suggested by the reviewer, in addition to Figure 6 showing the comparison between the modelling results and the in-situ observations, a new figure (Fig.S7) has been added in the Supplementary to show more the details of the comparison. In this revision, in addition to the evaluation based on correlation coefficient, the RMSE of the simulation against the in-situ observation is added in L220-227 according to the reviewer's comments, and presented below:

*Compared against the in-situ observations, the RMSE of the simulated temperature is around 2.0 ℃ for the 45 lakes listed in Table S1. It is noticed that the bias in the simulation mainly due to its underestimation for the warmer seasons. Taking Ngoring Lake for example, for the season when the in-situ observed temperature is above 10℃, the RMSE could reach to 2.42 ℃ though $R^2$ is higher than 0.7. The bias of the simulated water temperature could be reduced or corrected if the model is calibrated against the observations. However, it is important to note that the simulated water temperature is not completely equivalent to the in-situ observations. This is because the simulations represent the lake-wide mean temperature of the skin layer, while the in-situ observation used herein is the profile mean temperature for a fixed location in the lake.*